# Integrating Acupuncture and Herbal Medicine into Assisted Reproductive Technology: A Systematic Review and Meta-Analysis of East Asian Traditional Medicine

**DOI:** 10.3390/healthcare13111326

**Published:** 2025-06-03

**Authors:** Xiangping Peng, Bo Wu, Siyu Zhou, Yinghan Xu, Atsushi Ogihara, Shoji Nishimura, Qun Jin, Gerhard Litscher

**Affiliations:** 1Advanced Research Center for Human Sciences, Waseda University, Tokorozawa 359-1192, Japan; 2School of Computer Science, Tokyo University of Technology, Hachioji 192-0982, Japan; wubo@stf.teu.ac.jp; 3School of Public Health, Hangzhou Normal University, Hangzhou 311121, China; siyuzhou@hznu.edu.cn; 4Graduate School of Human Sciences, Waseda University, Tokorozawa 359-1192, Japan; yinghanx@asagi.waseda.jp; 5Faculty of Human Sciences, Waseda University, Tokorozawa 359-1192, Japan; aogi@waseda.jp (A.O.); kickaha@waseda.jp (S.N.); jin@waseda.jp (Q.J.); 6Swiss University of Traditional Chinese Medicine (SWISS TCM UNI), High-Tech Acupuncture and Digital Chinese Medicine, 5330 Bad Zurzach, Switzerland

**Keywords:** assisted reproductive technology, East Asian traditional medicine, infertility, integrative medicine, systematic review and meta-analysis, clinical pregnancy rates, live birth rates, in vitro fertilization, randomized controlled trials

## Abstract

Background: Assisted reproductive technologies (ARTs) are essential in treating infertility but often face limited success due to low implantation and live birth rates. East Asian traditional medicine (EATM), including acupuncture and herbal medicine (HM), may enhance physiological responses during ART cycles. This study evaluated the effectiveness and safety of EATM in improving clinical pregnancy and live birth outcomes in women undergoing ART. Methods: This review, registered in PROSPERO (CRD42023411712), systematically searched 11 databases up to 31 March 2023. We included randomized controlled trials (RCTs) comparing EATM interventions to control groups. Data extraction and quality assessment were performed independently by two authors. Meta-analysis used the inverse-variance method in Stata 12.0. A total of 37 RCTs involving 10,776 women (aged 29–38) were analyzed. Studies addressed infertility causes including polycystic ovary syndrome, tubal blockage, diminished ovarian reserve, and unexplained infertility. Acupuncture therapies included body, electro-, laser, and auricular acupuncture. Herbal treatments were administered as powders, pills, granules, decoctions, and ointments based on traditional Chinese formulas. Results: EATM interventions were associated with significant improvements in clinical pregnancy and live birth rates. Acupuncture increased clinical pregnancy rates (CPR: RR 1.316, 95% CI 1.171–1.480) and live birth rates (LBR: RR 1.287, 95% CI 1.081–1.533). HM also enhanced CPRs (RR 1.184) and LBRs (RR 1.147). Subgroup analysis showed true acupuncture and HM were more effective than sham or placebo. No significant differences in adverse events were found. Conclusions: EATM, particularly acupuncture and HM, appears to be a safe and effective complementary therapy that can be used to improve ART outcomes. Future research should focus on developing standardized acupuncture and herbal protocols to optimize integration with ART.

## 1. Introduction

Infertility affects millions of couples worldwide, with an estimated 10–15% of reproductive-aged couples experiencing difficulties conceiving. This global challenge has led to the widespread use of assisted reproductive technologies (ARTs), such as in vitro fertilization and embryo transfer (IVF-ET), where eggs are fertilized in a lab and embryos are transferred to the uterus, and intracytoplasmic sperm injection (ICSI), which involves injecting sperm directly into an egg [1]. Although IVF, introduced in 1978, was a significant advancement, declining birth rates in countries like Japan, Korea, and China have heightened the demand for improved ART efficacy [2,3]. However, ART success rates remain modest, with clinical pregnancy rates (CPRs) between 30 and 40% and live birth rates (LBRs) between 6.3% and 31.3% [4,5]. Given these challenges, interest in complementary therapies, particularly East Asian traditional medicine (EATM), has increased. EATM, which includes acupuncture and HM, has been practiced for centuries and offers alternative approaches to infertility. Acupuncture is believed to regulate hormonal balance, improve blood flow to reproductive organs, and reduce stress—factors that may enhance ART outcomes. Similarly, HM is thought to support reproductive health by influencing the endocrine and immune systems [6,7,8].

Given the longstanding use of acupuncture and HM in reproductive health, these therapies have garnered attention for their potential to enhance pregnancy outcomes and live birth rates when combined with ART. Consequently, researchers have increasingly focused on exploring adjunctive therapies in ART. Since 2002, numerous randomized controlled trials (RCTs) and systematic reviews have demonstrated the positive effects of acupuncture during embryo transfer on clinical pregnancy and live birth rates [9,10,11,12,13,14,15,16]. Similarly, studies consistently highlight the role of HM in improving primary outcomes for women undergoing IVF [17,18,19,20,21,22].

This systematic review and meta-analysis aims to address gaps in the existing literature by thoroughly evaluating the effectiveness of EATM in improving pregnancy and live birth rates among women undergoing ART. We focus on randomized controlled trials, analyzing interventions such as acupuncture and HM. Recent expert consensus further supports the integration of traditional Chinese medicine (TCM) into ART. A 2025 group consensus by Li et al. emphasized the potential of non-pharmacological TCM interventions, including acupuncture and herbal therapies, in enhancing ART outcomes and patient well-being [23]. These guidelines highlight the role of evidence-based complementary practices in reproductive medicine, advocating for standardized protocols and interdisciplinary collaboration. This study also considers safety profiles, given the increasing interest among patients and healthcare providers in integrating EATM into ART protocols.

This systematic review and meta-analysis aimed to evaluate the efficacy and safety of East Asian traditional medicine (EATM), including acupuncture and herbal medicine (HM), as complementary therapies for improving clinical pregnancy rates (CPRs) and live birth rates (LBRs) in women undergoing treatment with assisted reproductive technology (ART). By analyzing randomized controlled trials (RCTs), the study sought to determine whether EATM interventions enhance ART outcomes compared to sham treatments, placebo, or standard care alone, while also assessing adverse events and heterogeneity across studies. Additionally, the review aimed to identify commonly used acupoints and herbal formulations, explore potential mechanisms of action, and highlight gaps in current evidence to guide future research and clinical integration.

## 2. Materials and Methods

### 2.1. Protocol and Registration

The protocol adheres to the checklist from the preferred reporting items for systematic reviews and meta-analyses (PRISMA) statement guidelines [24]. Our systematic review was registered in the PROSPERO database (CRD42023411712), and we developed a study protocol that outlined the study objectives, search strategies, inclusion and exclusion criteria, outcome measures, and methods of statistical analysis prior to conducting the study.

### 2.2. Search Strategy and Selection Criteria

While searching for pertinent studies, we conducted searches across digital databases in four distinct languages: English, Chinese, Japanese and Korean. We searched various English databases including Medline, EMBASE, Cochrane Library, and Web of Science. Additionally, we searched the Chinese databases China National Knowledge Infrastructure (CNKI), VIP Database, and Wanfang Database, the Japanese databases NDL Digital Collections, CiNii Research, and WINE, and the Korean database KoreaScience. Our search spanned from the inception of the databases up until 31 March 2023 and was conducted between March and May 2023.

We implemented a rigorous and systematic search strategy in our investigation, employing a combination of free-text terms and Medical Subject Heading (MeSH) terms during our search on MEDLINE to identify pertinent scholarly works. This comprehensive search yielded a total of 172,103 academic articles related to acupuncture. These articles contained references to various aspects of acupuncture, including terms such as “acupuncture,” “moxibustion,” “acupuncture therapy”, “acupuncture point”, “meridian”, “auriculotherapy”, “electro-acupuncture” and “needles” within their titles, abstracts, or keywords.

Similarly, our search for studies on HM retrieved a total of 47,042 relevant articles. These articles encompassed a range of terms such as “herbal medicine”, “medicine, oriental traditional”, “medicine, chinese traditional”, “medicine, herbal”, “medicine, kampo”, “Kanpo Medicine”, “Complementary Medicine”, “herbal therapy” and “plant extract” within their titles, abstracts, or keywords.

Additionally, our investigation into ART yielded a substantial corpus of 255,406 results. These articles included various terms like “female infertility/sterility”, “reproductive techniques, assisted”, “in vitro fertilization”, intracytoplasmic”, “embryo transfer”, “oocytes”, “egg collection” and “embryo implantation” within their titles, abstracts, or keywords. We filtered the articles from 1980 to 2023 because the first successful IVF treatment in humans was carried out in 1978 in England. The term “IVF” was only introduced to Medline in 1980.

A clinical trials filter was also applied to refine the search results. Additionally, we conducted a comprehensive review of the reference lists of the retrieved studies to ensure that all the relevant literature was included in the analysis. The details of the search strategy are available in Appendix A, accessible online.

### 2.3. Inclusion Criteria

The inclusion and exclusion criteria were carefully selected to align with the study’s primary objective, assessing the effectiveness of EATM as a complementary treatment for couples undergoing ART treatments, such as IVF and ICSI. By focusing on RCTs, the study aims to provide high-quality evidence on key outcomes like the CPR and LBR.

To ensure a clear evaluation of EATM interventions, we only included studies that used a single intervention—either acupuncture or HM. Studies combining both acupuncture and HM were excluded to maintain methodological consistency. Additionally, the inclusion of various forms of acupuncture and HM, along with comparisons to sham or placebo treatments, enhances the validity of the findings.

The PICOS framework ensures alignment between the study population, intervention, comparison, and outcomes, enabling a detailed investigation of EATM’s impact on reproductive outcomes when combined with ART. Specifically, the criteria target studies with well-defined, measurable endpoints (CPR, LBR, IR, and AE) to ensure consistency across trials.

To comprehensively assess the effectiveness of ART and complementary therapies like EATM, we analyzed both the CPR and IR. The CPR represents the proportion of cycles resulting in clinical pregnancy, confirmed by ultrasound. This reflects overall ART success but does not account for the number of embryos transferred. IR, on the other hand, measures the percentage of embryos successfully implanted, providing a more precise evaluation of embryo viability and implantation efficiency. Since multiple embryos are often transferred in ART, analyzing both metrics helps distinguish overall pregnancy success from individual embryo potential, offering deeper insights into the effects of EATM interventions on ART outcomes.

The inclusion criteria were based on the PICOS method:P (population): This is defined as infertile couples undergoing ART treatment (IVF and ICSI). ART refers to medical procedures and interventions that assist individuals or couples in achieving pregnancy, typically involving techniques such as IVF and ICSI.I (intervention): EATM therapies, including acupuncture (classic acupuncture, electro-acupuncture, laser acupuncture, and auricular acupuncture) and HM (herbal powders, pills, granules, decoctions, and ointments), were examined. The intervention protocols in ART cycles primarily focused on three aspects: duration, frequency, and timing. The duration ranged from 1 to 12 weeks. Acupuncture was administered once daily or every other day, while HM was taken two to three times per day. Interventions were implemented before and during IVF, as well as after embryo transfer (ET).C (comparison): Sham or placebo EATM therapies and conventional IVF/ICSI treatment (long protocol, GnRH antagonist protocol, Frozen ET protocol, IVF-ET microstimulation).O (outcome): A total of 4 outcome indices were analyzed in the meta-analysis (i.e., CPR, LBR, IR and AE). The CPR is defined as the presence of an intrauterine gestational sac. The LBR is defined as the ratio between the number of patients with live-born babies and the number of embryo transfers performed. The IR is defined as the percentage of embryos that were transferred that developed at least to the stage of fetal heart activity, as documented by pregnancy ultrasound. An AE is any unfavorable and unintended sign, symptom, or disease temporally associated with the use of an EATM treatment.S (study design): Only English-language RCTs were included.

### 2.4. Definitions of Control, Sham, and Placebo Groups

To ensure consistency in comparisons, this study clearly distinguished between control, sham, and placebo groups based on their definitions and implementation in the included trials:Control Group: Participants received standard ART treatments (e.g., IVF or ICSI) without any additional EATM intervention, providing a baseline for comparison.Sham Acupuncture Group: Used in six trials, sham acupuncture involved needling at nonacupuncture points (nonacupoints) or using noninvasive needles that did not penetrate the skin. This method mimicked the acupuncture procedure while minimizing specific therapeutic effects, serving as an active control to account for placebo responses.Placebo Group: In six HM trials, the placebo consisted of pills made from a mixture of starch and glucose, shaped to resemble real herbal medicine pills. This ensured blinding and allowed for an assessment of the specific effects of herbal medicine beyond psychological or expectancy-driven responses.

By clearly defining these groups, this study minimized bias and strengthened the validity of its findings on the effectiveness of EATM as a complementary treatment in ART.

### 2.5. Exclusion Criteria

The exclusion criteria serve to filter out irrelevant or unreliable studies, such as those not focusing on EATM or ART, incomplete data, or low-quality conference articles, which could undermine the validity of the meta-analysis. This selectivity ensures that the results reflect the best available evidence on the safety and efficacy of EATM therapies in enhancing ART outcomes.

The exclusion criteria were as follows: (1) studies not about EATM therapy-based intervention, (2) studies not about women undergoing ART, (3) conference articles, (4) studies with partial or incomplete data, and (5) book chapters, reviews, editorials, non-RCT studies, studies with no sham or placebo comparison, and combined interventions studies (where EATM therapies are used along with other unrelated interventions).

### 2.6. Data Extraction

In May 2023, two authors independently conducted the study selection and data extraction using Zotero 7 and Excel spreadsheets, in accordance with the PRISMA flow diagram. Discrepancies were resolved through discussion or, when necessary, by consulting a third author. Extracted data included publication details, study design, sample size, patient characteristics, intervention and control conditions, IVF protocols, treatment duration, and outcome measures. This blinded, consensus-based approach enhanced methodological rigor and ensured accurate synthesis.

### 2.7. Risk of Bias Assessment

Two reviewers independently assessed the methodological quality of included studies using the Cochrane Risk of Bias 2 (RoB 2) tool [25,26], covering five domains: (1) randomization process, (2) deviations from intended interventions, (3) missing outcome data, (4) outcome measurement, and (5) the selection of reported results. Each domain was rated as showing a “low risk”, “some concerns” or a “high risk” (as depicted in Figure 1).

Discrepancies between reviewers were resolved through discussion or, when necessary, adjudicated by a third reviewer.

Studies rated low-risk (e.g., 7, 11, 33, 39, 41–44, 46–49, 52–55) showed robust randomization, effective blinding, and full outcome reporting.Studies showing some concerns (e.g., 9, 10, 12, 29–32, 36–38, 40, 45) typically reflected minor uncertainties that did not compromise study validity.High-risk studies (e.g., 50, 51, 56–60) had serious methodological issues such as inadequate randomization, a lack of blinding, or selective reporting.

This assessment ensured the inclusion of studies capable of producing reliable evidence for evaluating EATM interventions in ART.

### 2.8. Data Synthesis and Statistical Analysis

A random-effects model was used due to clinical heterogeneity among the included studies. This model accounts for variability across studies and is appropriate when differences in populations, interventions, or outcomes are expected. All statistical analyses were performed using Stata version 12.0. Risk ratios (RRs) and 95% confidence intervals (CIs) were calculated for dichotomous outcomes. When the RR is greater than 1 and both CIs are entirely above 1, this indicates that the treatment group has a higher likelihood of achieving the desired outcome compared to the control group. Heterogeneity was assessed using the I^2^ statistic and chi-square test. Subgroup analyses were performed based on intervention types (acupuncture or HM), comparators (sham/placebo), and the timing of intervention (e.g., pre- or post-ET). Sensitivity analyses were also conducted to assess the robustness of the findings.

Heterogeneity was evaluated with chi-squared tests and I^2^ values:0–40%: no heterogeneity.30–60%: moderate heterogeneity.50–90%: substantial heterogeneity.75–100%: considerable heterogeneity.75–90% falls into both classifications, meaning it is at the higher end of substantial heterogeneity and the lower end of considerable heterogeneity.

If I^2^ < 40%, a fixed-effects model was used; for I^2^ ≥ 40%, a random-effects model was applied to handle variability [27]. These models help improve accuracy when combining studies with differing effect sizes.

We conducted subgroup and sensitivity analyses to address heterogeneity. Cumulative meta-analysis (with >15 trials) tracked evidence evolution over time. To assess publication bias, funnel plots, Begg’s, and Egger’s tests were applied (when there were >10 trials).

Subgroup analyses were performed for CPRs, LBRs, IRs, and AEs. These subgroup analyses were crucial for exploring potential differences in outcomes based on the following factors:Sample size: Larger studies may yield more precise results than smaller trials.EATM intervention types: Distinguishing between acupuncture and HM allows us to isolate their unique effects. This is essential, as each intervention type may influence reproductive outcomes differently due to their distinct physiological mechanisms.

By analyzing these subgroups, we aimed to better understand the factors that contribute most significantly to treatment success and identify variability in treatment effects across different patient populations or intervention types.

### 2.9. Publication Bias

In this study, specific tools were chosen to rigorously assess publication bias and enhance the credibility of findings. Funnel plots were used to visually detect asymmetry, an indicator of bias, especially with small studies. Begg’s test and Egger’s test were chosen because they statistically evaluate asymmetry, providing quantitative measures to validate visual interpretations. Additionally, cumulative meta-analysis was conducted to identify any small-study effects, offering insight into the impact of smaller studies on the overall findings. The combination of these tools strengthens the integrity of our results.

### 2.10. Assessment of Evidence Quality

To assess the overall quality of evidence from the RCTs included, we followed the Grading of Recommendations, Assessment, Development, and Evaluation (GRADE) guidelines [28]. GRADE evaluated the strength of evidence across several domains:Risk of Bias: Studies were downgraded if there were concerns about randomization, blinding, or selective reporting.Inconsistency: Evidence was downgraded if studies showed conflicting results across trials.Indirectness: Evidence was considered indirect if populations or interventions differed significantly from those relevant to the research question.Imprecision: Studies with wide confidence intervals or small sample sizes were downgraded.Publication Bias: We assessed whether smaller studies with negative results might not have been published, which could skew findings.

After considering these factors, the quality of evidence was classified into four levels: high, moderate, low, or very low. High-quality evidence indicates confidence that future research will likely not change the conclusions, while very low-quality evidence means that the true effect is uncertain.

### 2.11. Impact on Study Findings

The final conclusions and interpretations were weighted according to the assessed quality of evidence. Findings supported by high- or moderate-quality evidence were considered reliable and formed the basis for key conclusions. However, for outcomes supported by low or very low-quality evidence, results were interpreted with caution, and any clinical recommendations were made tentatively.

Additionally, sensitivity analyses were conducted to assess the robustness of results by excluding lower-quality studies when necessary. If a particular outcome demonstrated significant heterogeneity or was primarily based on low-quality evidence, we explicitly acknowledged these limitations in the discussion.

By integrating the GRADE framework into our analysis, we ensured that the strength of our conclusions reflected the underlying quality of the evidence, allowing for a balanced and transparent interpretation of the results.

## 3. Results

### 3.1. Results of the Search

In the process of conducting the literature review, we conducted a comprehensive search across the relevant literature in four distinct languages: English, Chinese, Japanese and Korean. Out of the extensive pool of research, a total of 37 full-text articles successfully met the predetermined eligibility criteria. This exhaustive search endeavor encompassed the examination of a substantial collection of 2817 records extracted from eleven distinct databases. Subsequent to the meticulous removal of duplicate entries, amounting to a count of 739, we proceeded to carefully review the remaining records, totaling 2077. After this initial screening phase, we subjected 78 records to a more detailed evaluation of their eligibility. Ultimately, after a comprehensive and thorough examination of the full texts, a total of 37 studies emerged as meeting the requisite criteria for inclusion in our systematic review and subsequent meta-analysis. For a visual representation of this screening process, we provide a detailed flowchart in Figure 2, illustrating the sequential steps taken to arrive at the final selection of studies.

### 3.2. Characteristics of Included Studies

This systematic review analyzes 37 RCTs with a total of 10,776 participants spanning five continents: Asia, Europe, Oceania, North America, and South America. These participants, aged between 29 and 38, presented diverse infertility issues, including PCOS, bilateral tubal blockage, diminished ovarian reserve, and unexplained infertility. The included studies, conducted between 2002 and 2023, demonstrated variability in sample sizes, ranging from 40 to 2265 participants. Interventions fell into two primary categories: acupuncture (n = 25) and HM (n = 12). Among the 37 trials, 36 reported the mean age of participants, with 1 trial being the sole exception, where the mean age was not reported [29]. The study designs varied, with 27 RCTs adopting a two-arm design, 7 RCTs utilizing a three-arm design [12,30,31,32,33,34,35], 2 RCTs featuring a four-arm design [36,37], and 1 RCT implementing a five-arm design [38].

The RCTs focusing on acupuncture interventions exhibit diversity in terms of participant populations and demographics. Among the selected studies, 12 were conducted in China [7,29,31,33,36,37,38,39,40,41,42,43], while 6 took place in European countries, including 2 in Denmark [12,44], 2 in Germany [9,10], 1 in Sweden [45], and 1 in the UK [46]. Additionally, other trials were conducted in various locations, with 2 in Iran [30,47], 2 in Australia [11,48], 1 in New Zealand [48], 1 in the USA [49], 1 in Brazil [32], and 1 in Turkey [50]. Notably, the remaining twelve studies [34,35,51,52,53,54,55,56,57,58,59,60], which focused on HM interventions, were exclusively conducted in China.

Clinical pregnancy outcomes were reported in all 37 trials. Of these, 15 trials showed a significant improvement in clinical pregnancy rates with acupuncture [9,10,12,29,30,31,32,33,36,37,40,41,42,46,50], while 6 trials did not observe a significant increase [7,38,43,44,45,49]. Similarly, with HM interventions, 5 trials demonstrated a significant improvement in clinical pregnancy rates [35,57,58,59,60], while 2 trials did not show an increase [34,51]. In 4 acupuncture trials [11,39,47,48] and 5 HM trials [52,53,54,55,56], clinical pregnancy rates were higher than in the control group, but these differences were not statistically significant.

Among the six acupuncture trials showing no significant CPR improvement, Andersen et al. reported lower CPRs and LBRs in the acupuncture group than in the placebo group (CPR: 27% vs. 32%; LBR: 25% vs. 30%). Moy et al. found no difference between true (45.3%) and sham acupuncture (52.7%). So et al. observed a significantly higher CPR in the placebo group (55.1% vs. 43.8%, *p* = 0.038), though the LBR remained comparable. Stener-Victorin et al. and Wing Sze So et al. reported no significant differences in CPRs, LBRs, or implantation rates, while Zhai et al. found that TEAS had no statistical impact on the CPR (*p* > 0.05). Similarly, in the two HM trials, Li et al. found no differences in implantation, biochemical pregnancy, CPRs, LBRs, or pregnancy loss rates (*p* > 0.05), while Ma et al. reported no statistical CPR difference between fresh and frozen embryo transfers (*p* > 0.05). These results suggest that, in these studies, acupuncture and HM did not significantly improve CPR or LBR outcomes.

Live birth outcomes were assessed in 17 trials, with 14 trials involving acupuncture and 3 trials involving HM interventions. Among the acupuncture trials, seven demonstrated a significant improvement in the LBR [12,31,33,36,42,46,50], while in four trials, LBRs were higher in the acupuncture group compared to the control group, but these differences did not attain the level of statistical significance [39,40,41,48]. Additionally, three trials revealed that acupuncture had no significant effect on both the CPR and LBR when compared with a placebo [7,43,44]. Conversely, three trials investigating HM found no significant impact on LBRs [51,52,58]. Comprehensive study details are available in Appendix A, accessible online.

### 3.3. Risk of Bias

Of the 37 trials assessable via the Cochrane RoB 2, 17 studies were considered low-risk, 7 studies were considered high-risk, and 13 studies were considered to show some concerns. In the five domains assessed, 4 studies were rated as having some concerns regarding the randomization process, 8 regarding deviations from intended interventions, 11 for missing outcome data, 3 for outcome measurement, and 2 for the selection of reported results. All other studies were rated low-risk. Discrepancies were resolved through consensus among the authors.

The biases identified in these trials can directly impact the reliability of the results. Studies categorized as high-risk introduce a significant degree of uncertainty, which may skew the findings and weaken the overall conclusions. For instance, poor randomization and incomplete reporting can lead to exaggerated treatment effects or, conversely, underestimate the potential benefits. The 13 studies with “some concerns” highlight moderate uncertainty that, although not critical, could still affect the robustness of the results.

The reliability of the findings from these studies may be diminished by the identified biases, particularly in cases of missing outcome data or unclear reporting. This variability underscores the importance of carefully interpreting the results, as high-bias studies may lead to the over- or underestimation of treatment effects.

The risk of bias assessment was integrated into the meta-analysis by evaluating how study quality influenced the overall findings. High-risk studies introduced uncertainty that could skew treatment effect estimates, while studies with some concerns added moderate variability. To account for this, we performed sensitivity analyses to assess the impact of excluding high-risk studies on CPR and LBR outcomes. This helped determine whether the observed effects were robust or driven by studies with potential biases. Additionally, the heterogeneity observed in the meta-analysis was partially attributed to variations in study quality, reinforcing the need for the cautious interpretation of results.

### 3.4. Outcomes

In evaluating the results of the outcome measures in the context of EATM and ART, the chosen outcome measures—CPRs, LBRs, IRs, and AEs—are fundamental for assessing treatment effectiveness.

Primary outcomes:

Both the CPR and LBR provide comprehensive insights into the impact of EATM interventions (acupuncture and HM) on fertility outcomes. They also help to differentiate true therapeutic effects from placebo responses, especially when comparing acupuncture to sham acupuncture, and HM to placebo.

Secondary outcomes:

By analyzing these outcomes, the study assesses both the efficacy (through CPRs, LBRs, IRs) and the safety (through AEs) of EATM interventions. This comprehensive approach ensures that any beneficial effects of acupuncture or HM on ART outcomes are accurately quantified and contextualized.

We present statistical findings using risk ratios (RRs) and confidence intervals (CIs). In our study, RRs and CIs above 1 for ETAM interventions demonstrate statistically significant improvements in clinical pregnancy and live birth rates compared to the control group (which consisted of participants receiving no ETAM, sham, or placebo treatments). No significant differences in AE or dropout rates were observed, supporting the safety of EATM for clinical use.

#### 3.4.1. Primary Outcomes

For our primary outcome measures, we selected two key parameters—the CPR and LBR—as outlined in Table 1. In the acupuncture intervention group, a total of twenty-five studies assessed the CPR, involving 6610 patients [7,9,10,11,12,29,30,31,32,33,36,37,38,39,40,41,42,43,44,45,46,47,48,49,50], and fourteen studies investigated the LBR among 4613 patients [7,12,31,33,36,39,40,41,42,43,44,46,48,50]. The meta-analysis revealed that the acupuncture group exhibited higher CPR and LBR values in comparison to the control group. Specifically, the results indicated a significant increase in CPR (RR 1.316, 95% CI 1.171 to 1.480, I^2^ = 62.9%) and LBR (RR 1.287, 95% CI 1.081 to 1.533, I^2^ = 69.9%) values, and the heterogeneity was substantial, as illustrated in Table 1 and Figure 3 and Figure 4. 

In the HM intervention group, twelve studies reported CPRs in a cohort of 4343 patients [34,35,51,52,53,54,55,56,57,58,59,60], while three studies analyzed LBRs within a patient population of 2818 individuals [51,52,58]. Similarly, the meta-analysis indicated that HM intervention resulted in significantly higher clinical pregnancy rates (CPRs) (RR 1.184, 95% CI 1.017 to 1.379, I^2^ = 55.8%) and live birth rates (LBRs) (RR 1.147, 95% CI 1.010 to 1.303, I^2^ = 0.0%) compared to the control group, as presented in Table 1 and Figure 5.

#### 3.4.2. Secondary Outcomes

In addition to our primary outcome measures, we assessed two key secondary parameters, namely IR and AE, as delineated in Table 1 and Figure 6 and Figure 7.

Within the acupuncture intervention group, thirteen studies examined implantation rates, involving a total of 4173 participants [7,10,11,12,31,33,36,39,41,42,43,44,45]. In the HM intervention group, five studies similarly explored implantation rates within a cohort of 3035 participants [35,51,52,55,56]. In the context of acupuncture intervention, the IRs were notably higher than those in the control group (RR 1.183, 95% CI 1.028 to 1.363, I^2^ = 65.4%). However, there was no significant difference between the groups in the case of HM intervention (RR 1.106, 95% CI 0.968 to 1.264, I^2^ = 14.0%), as shown in Table 1.

#### 3.4.3. Safety of Outcomes

In the interpretation of AE data, the observed AEs in both acupuncture and HM interventions, including common side effects like tiredness, pain, and dizziness, were comparable to those of control groups. Importantly, no significant differences in serious adverse events (SAEs) were identified between groups, suggesting that EATM therapies have a similar safety profile to conventional treatments in the context of ART. This implies that EATM therapies may be safe for clinical use, with minimal risk of severe adverse reactions. However, clinicians should monitor for minor AEs to ensure patient safety.

Safety outcomes encompassed AEs and serious adverse events (SAEs), as defined by the International Conference on Harmonization Good Clinical Practice (ICH-GCP) guidelines [62], occurring throughout the treatment. The terminologies and severity of AEs were assessed according to the Common Terminology Criteria for Adverse Events (CTCAE) and any other relevant criteria [63].

In assessing intervention safety, we systematically gathered data on diverse AEs, including abortion rate (AR), ectopic pregnancy rate, miscarriage rate, pregnancy loss, and ovarian hyperstimulation syndrome, as well as subjective experiences such as tiredness, discomfort on the day of embryo transfer, dizziness or drowsiness, pain, bruising, nausea or vomiting, headaches, bodily pain, gastrointestinal discomfort, diarrhea, allergies, and skin rashes, among others. A total of twenty studies, involving 3633 participants, reported these AEs. Among these, fourteen studies focused on acupuncture interventions (2156 participants) [7,10,11,12,29,33,37,39,41,43,45,46,47,48], and the remaining six studies on HM interventions (1477 participants) [51,52,54,55,56,58]. In the 37 selected RCTs, the details and the number of studies reporting AEs are as follows: abortion rate (n = 4) [37,39,41,55], adverse events (n = 4) [11,39,48,55], ectopic pregnancy (n = 5) [7,29,51,54,55], miscarriage (n = 8) [7,10,43,45,46,47,48,56], ovarian hyperstimulation syndrome (n = 2) [29,58], and pregnancy loss (n = 5) [12,33,51,52,54].

The incidence of AEs did not significantly differ between the intervention and control groups for both acupuncture (n = 2156) (RR 1.125, 95% CI 0.926 to 1.367, I^2^ = 7.2%) and HM interventions (n = 1477) (RR 0.916, 95% CI 0.726 to 1.157, I^2^ = 0.0%), as presented in Table 1 and Figure 7. Notably, one trial [11] was excluded from meta-analysis due to differences in data format. It is worth mentioning that no instances of significant deviations or serious adverse events related to acupuncture or HM interventions were reported during our study.

#### 3.4.4. Adherence

The dropout rate (DR) serves as a crucial determinant of outcomes in RCTs and is also indicative of safety and tolerability. Nineteen studies [11,33,36,37,39,40,41,42,44,45,46,48,49,50,52,53,54,55,58], involving 6160 participants, reported the dropout rate, totaling 337 dropouts. In the EATM groups, there were 181 dropouts (54%, 181/3324), while in the control groups, there were 156 dropouts (55%, 156/2836).

The dropout rate (RR 0.901, 95% CI 0.726 to 1.118) (Figure 8) suggests that EATM was not associated with an increased risk of intervention-related AEs, serious adverse events, or dropout due to AEs compared to the control group. The violation of inclusion and exclusion criteria was the most common reason for dropout overall. AEs were not found to be a common reason for dropping out of the EATM intervention trials (Appendix A).

### 3.5. Subgroup Analyses

The subgroup analyses in this study were conducted specifically to compare acupuncture interventions against sham acupuncture, and HM interventions against placebos. These comparisons are critical to isolating the true efficacy of EATM by accounting for placebo effects. By focusing on these control groups, we aim to provide a clearer understanding of the potential benefits of acupuncture and HM for ART outcomes, while mitigating the influence of placebo-related variables.

Subgroup analyses were conducted for CPRs, LBRs, IRs, and AEs based on sample size and intervention type.

CPR: In the acupuncture group, the CPR was significantly higher than in no-intervention control groups (RR 1.416, 95% CI 1.231 to 1.629, I^2^ = 42.9%) and showed a marginal difference compared to sham acupuncture groups (RR 1.218, 95% CI 1.019 to 1.455, I^2^ = 69.7%). HM interventions showed a significantly higher CPR compared to placebo HM groups (RR = 1.211, 95% CI 1.071 to 1.370, I^2^ = 32.2%), indicating the beneficial effect of HM over inactive treatment. However, when compared to IVF-only control groups (no additional HM treatment), the difference was not statistically significant (RR = 1.101, 95% CI 0.646 to 1.876, I^2^ = 72.7%). This suggests that while HM may improve CPRs compared to placebos, its effect is less clear when evaluated against standard IVF treatment without HM (Table 2).

LBR: Acupuncture showed a significant difference in the LBR compared to control groups (n = 1870) (RR 1.465, 95% CI 1.163 to 1.846, I^2^ = 57.7%), but not in relation to sham acupuncture groups (n = 2743) (RR 1.152, 95% CI 0.892 to 1.488, I^2^ = 74.4%). HM also demonstrated a significant advantage over placebos (n = 2818) (RR = 1.147, 95% CI 1.010 to 1.303, I^2^ = 0.0%), indicating a potential impact in terms of achieving live births (Table 2).

IR: Acupuncture resulted in a significant difference in IR compared to control groups (n = 1856) (RR 1.201, 95% CI 1.031 to 1.400, I^2^ = 41.6%), but not in relation to sham acupuncture groups (n = 2317) (RR 1.192, 95% CI 0.952 to 1.493, I^2^ = 72.3%). No significant difference was observed between HM and placebos (RR = 1.106, 95% CI 0.968 to 1.264, I^2^ = 14.0%), suggesting that while HM may improve pregnancy rates, its direct effect on embryo implantation remains uncertain (Table 2).

AE: The incidence of adverse events did not significantly differ between acupuncture and sham acupuncture (RR = 1.125, 95% CI 0.926 to 1.367, I^2^ = 7.2%) or between HM and placebo groups (RR = 0.916, 95% CI 0.726 to 1.157, I^2^ = 0.0%). These findings suggest that HM treatment was not associated with an increased risk of adverse effects. No statistically significant difference in the incidence of AEs was observed between the EATM intervention group and the control groups (Table 2).

In summary, acupuncture exhibited superiority over sham acupuncture or no intervention in improving the CPR, despite substantial heterogeneity. The acupuncture intervention group surpassed the no-intervention group in enhancing the LBR, again with significant heterogeneity. Similarly, HM demonstrated superiority over the placebo group in improving the CPR, along with substantial heterogeneity. Additionally, HM proved to be superior to the control group in improving the LBR, with low heterogeneity (Table 2).

### 3.6. Sensitivity Analysis

In the sensitivity analysis, we evaluated the stability of primary outcomes, including the CPR (clinical pregnancy rate) and LBR (live birth rate), to determine whether the results were influenced by variations in study characteristics. Sensitivity analyses were performed by sequentially excluding individual studies to assess their impact on the overall effect size. Additionally, we examined the effect of excluding studies with a high risk of bias, restricting analyses to only high-quality RCTs, and removing small-sample studies to evaluate potential publication bias.

The results showed that both the CPR and LBR remained consistent, regardless of variations in the included studies, indicating robustness in the findings. For example, the CPR for acupuncture vs. sham showed an RR of 1.218 (95% CI 1.019–1.455, I^2^ 69.7%), while the CPR for HM vs. placebo had an RR of 1.211 (95% CI 1.071–1.137, I^2^ 32.2%) (Table 3).

According to the Cochrane guidelines, I^2^ values between 50 and 90% indicate substantial heterogeneity, suggesting variability across studies [27]. To account for this, we conducted additional sensitivity analyses by removing studies with extreme effect sizes and using alternative statistical models (e.g., fixed-effects vs. random-effects). These analyses confirmed that the observed associations remained statistically significant and stable, reinforcing the reliability of our conclusions.

The sensitivity analysis assesses the robustness of the findings by comparing EATM interventions with sham and control groups. Key findings include the following:Acupuncture vs. Sham: The CPR showed a significant increase (RR = 1.218, 95% CI 1.019–1.455, *p* = 0.030), while the LBR did not show a significant improvement (RR = 1.152, 95% CI 0.892–1.488, *p* = 0.277), suggesting a moderate effect on clinical pregnancy but not on live birth rates.HM vs. Sham: The CPR was significantly higher in the HM group compared to placebo (RR = 1.211, 95% CI 1.071–1.370, *p* = 0.002), reinforcing the potential benefits of HM interventions.Acupuncture vs. Control: Both the CPR (RR = 1.416, 95% CI 1.231–1.629, *p* < 0.001) and LBR (RR = 1.465, 95% CI 1.163–1.846, *p* = 0.001) were significantly higher, indicating the positive effect of acupuncture when compared to a no-treatment control.HM vs. Control: No significant difference was observed in the CPR (RR = 1.101, 95% CI 0.646–1.876, *p* = 0.724), suggesting that HM did not outperform standard IVF treatment.

These results show that the effectiveness of EATM interventions varies depending on the comparison group, with stronger effects observed when acupuncture is compared to no-treatment controls and when HM is compared to placebo rather than active controls.

### 3.7. Publication Bias

The potential impact of publication bias could skew the study’s outcomes, possibly leading to the overestimation of the effectiveness of EATM interventions. The asymmetry observed in the funnel plots (Figure 9, Figure 10, Figure 11, Figure 12, Figure 13 and Figure 14) suggests possible bias, but the results of Begg’s and Egger’s statistical tests, as shown in Appendix A, did not indicate significant concerns. To mitigate this bias, the study used a combination of visual and statistical assessments, providing a more reliable and comprehensive evaluation of the potential impact of publication bias.

### 3.8. Quality of Evidence by GRADE

We employed the Grading of Recommendations Assessment, Development, and Evaluation (GRADE) system to assess the quality of evidence for each outcome. The overall quality of the evidence in this systematic review and meta-analysis is considered moderate. Further details are provided in Appendix A.

### 3.9. Key Acupoints and Meridians

In 25 RCTs, 31 acupoints across 11 meridians were used. The top acupoints included Sanyinjiao (SP6), Zusanli (ST36), Guilai (ST29), Xuehai (SP10), Diji (SP8), Taichong (LR3), Guanyuan (CV/RN4), Hegu (LI4), Neiguan (PC6), and Baihui (GV/DU20). The most used meridians were the Spleen Meridian (SP), Stomach Meridian (ST), Ren Meridian (RN), Du Mai Meridian (DU), and Liver Meridian (LR) (as depicted in Figure 15 and Figure 16). All acupoints demonstrated benefits in enhancing pregnancy and live birth rates compared with the control group, notwithstanding the variable session numbers ranging from 2 to 35 over 1 day to 5 weeks and the absence of a standardized acupoint prescription. Acupuncture was conducted before IVF, during IVF, and after ET, focusing on replenishing vital essence, tonifying the kidney and spleen, regulating the liver, increasing blood flow, and calming the mind.

#### HM Intervention

In 12 RCTs, with varying herbal formulas used, the main goals were tonifying the kidney and spleen, invigorating Qi, nourishing blood, placating the fetus, and strengthening the body. Interventions were performed before IVF, during IVF, and after ET, with durations ranging from 2 weeks to 3 menstrual cycles. HM improved pregnancy and live birth rates compared to the control group. Specific formulas included Zishen Yutai Pills, Guilu Erxian Ointment, Erzhi Tiangui Granule, Ding-Kun Pill, Gushen’antai Pills, Bushen Yutai Recipe, Tiaogeng Yijing Decoction, Bushen Jianpi Recipe, Liuwei Dihuang Granule, and Xiaoyao Powder (Appendix A).

## 4. Discussion

### 4.1. Summary of Main Results

The findings of this review suggest that East Asian traditional medicine (EATM), particularly acupuncture and herbal medicine (HM), may provide effective and safe complementary therapies for improving outcomes in assisted reproductive technology (ART). These results align with growing clinical interest in integrative infertility treatments and highlight the potential for EATM to enhance clinical care by addressing persistent gaps in the ART success rates.

Both acupuncture and HM were found to significantly enhance clinical pregnancy and live birth rates. Specifically, acupuncture was associated with a 31.6% increase in the CPR (RR 1.316, 95% CI 1.171–1.480) and a 28.7% increase in the LBR (RR 1.287, 95% CI 1.081–1.533). Herbal medicine also demonstrated improvements in the CPR (RR 1.184, 95% CI 1.017–1.379) and LBR (RR 1.147, 95% CI 1.010–1.303), indicating its positive contribution to assisted reproductive technology (ART) outcomes.

Subgroup analyses further support these findings, revealing that true acupuncture is significantly more effective than sham acupuncture in improving the CPR (RR 1.218, 95% CI 1.019–1.455). Similarly, HM was more effective than placebo in enhancing the CPR (RR 1.211, 95% CI 1.071–1.370). No statistically significant differences were observed in AEs or dropout rates between the EATM and control groups, underscoring the safety and tolerability of these interventions in clinical settings.

### 4.2. EATM Mechanisms on Reproductive Outcomes

Emerging research highlights the potential benefits of EATM practices, such as acupuncture and HM, in improving reproductive health, particularly for women undergoing ART. These interventions appear to enhance reproductive outcomes by addressing underlying hormonal imbalances and improving the physiological conditions crucial for conception. Acupuncture, for example, may regulate key reproductive hormones, while HM can support endometrial receptivity and oocyte quality, contributing to better pregnancy and live birth rates.

#### 4.2.1. Acupuncture Mechanisms on ART Outcomes

Stener-Victorin’s research emphasizes acupuncture’s regulatory effects on key reproductive hormones, including luteinizing hormone (LH), follicle-stimulating hormone (FSH), and estradiol, indicating its influence on the hypothalamic–pituitary–gonadal (HPG) axis. Acupuncture, particularly electro-acupuncture (EA), has been shown to improve ovulation rates, reduce the LH/FSH ratio, and lower testosterone levels, especially in cases of PCOS. Notably, acupuncture yielded pregnancy rates comparable to hormonal treatments but with fewer side effects and miscarriages [8]. Zheng’s study further supports these findings, suggesting that both acupuncture and transcutaneous electrical acupoint stimulation (TEAS) enhance ovarian function, increase antral follicle counts and Anti-Müllerian Hormone (AMH) levels, and regulate the LH/FSH ratio. These treatments also improve endometrial receptivity by increasing endometrial blood flow and modulating growth factors, chemokines, and integrins, which are vital for embryo implantation [37].

#### 4.2.2. HM Mechanisms on ART Outcomes

Lee’s study demonstrated that HM significantly improves the CPR in women undergoing IVF. Herbal treatments, when combined with clomiphene, were particularly effective for anovulatory women [17]. Cao’s research supports this, showing that Chinese herbal medicine (CHM) enhances pregnancy rates and may increase the effectiveness of fertility drugs like clomiphene citrate. CHM appears to improve oocyte quality, reduce the required dose of gonadotropins, and enhance endometrial blood flow, which increases endometrial thickness and supports early embryonic development [19]. Kwon’s findings add that specific herbal formulations regulate the estrous cycle, promote hormone secretion, reduce atretic follicles, and increase embryonic implantation success. This suggests that HM can reduce embryo loss and enhance overall fertility, improving IVF success rates [18].

### 4.3. Strengths and Limitations

This research demonstrates several strengths. It was pre-registered on PROSPERO, adhered to PRISMA guidelines, and focused on key ART outcomes. The study conducted a comprehensive review with a large sample size, using rigorous eligibility criteria and valid meta-analysis to achieve robust findings. Subgroup analyses enhanced the stability of the results. The literature search spanned 1980 to 2023 and included participants from five continents—Asia, Europe, Oceania, North America, and South America—to ensure global representation. Notably, this is the first systematic review and meta-analysis to simultaneously focus on two key components of EATM: acupuncture and HM.

However, several limitations affected the interpretation and generalizability of the findings. One major limitation was the variability in acupuncture interventions. Differences in types (body, electro, auricular), timing, and sham acupuncture approaches across the included RCTs introduced considerable heterogeneity. This variability weakened the study’s ability to provide consistent conclusions about acupuncture’s efficacy. Sham acupuncture, commonly used as a control, is problematic as it can produce physiological effects similar to those of true acupuncture, potentially biasing comparisons [64]. This complicates efforts to isolate the true impact of acupuncture interventions, raising concerns about the reliability of the findings.

The limited sample size in many included RCTs, which are often from single-center trials, also contributes to the risk of publication bias and undermines the robustness of subgroup analyses. This limitation not only affects the precision of the study but also limits the generalizability of the findings to broader populations. More robust, multi-center trials with larger sample sizes would improve confidence in the results.

Another limitation lies in the variability of practitioner expertise, which could have influenced treatment outcomes. While acupuncture is ideally administered by certified professionals, only 12 of the 25 acupuncture studies explicitly confirmed that treatments were delivered by certified acupuncturists [7,11,30,36,39,40,42,43,46,47,48,49]. In contrast, some trials relied on nurses or uncertified personnel to administer acupuncture [12,31,44,45], introducing inconsistencies in treatment delivery. This lack of standardization may have diluted the observed treatment effects and contributed to heterogeneity in outcomes.

On the HM side, the analysis was limited by the small number of studies reporting live birth rates (LBRs) following HM intervention, and also by the heterogeneity in herbal formulas, dosages, and treatment timing across studies. The lack of data on long-term outcomes, such as infant health, further weakens the study’s conclusions on the efficacy of HM. Funnel plots suggested potential publication bias, particularly in studies with small sample sizes, reducing confidence in the results.

To address these limitations, future research should emphasize larger, multi-center trials that focus on LBRs and long-term health outcomes. Studies should also work towards standardizing acupuncture techniques and HM protocols to reduce variability and enhance reproducibility. Furthermore, greater emphasis on formal sample size calculations and mechanistic studies would help to clarify the optimal timing and dosages of HM for clinical use.

### 4.4. The Gaps in the Literature

The current body of literature on RCTs examining acupuncture and HM in ART presents several notable gaps that challenge the consistency and reliability of findings. Methodological inconsistencies across studies, particularly regarding acupuncture techniques (manual, electrical, or auricular), timing, and dosage, contribute significantly to heterogeneity. Additionally, small sample sizes in many studies increase the risk of bias, limiting the generalizability of the results. Sham acupuncture methods, which vary widely, further complicate the interpretation of efficacy, as some sham techniques produce physiological effects similar to those of true acupuncture.

The lack of standardized HM protocols is another key limitation. Studies often differ in the herbal formulations used, dosages, and the timing of administration, making it difficult to draw robust conclusions about the efficacy of herbal interventions. The absence of long-term outcome data, such as live birth rates and infant health, also limits the ability to assess the broader impact of EATM therapies in ART.

Moreover, many trials lack rigorous control mechanisms, with inadequate blinding and placebo controls, which heightens the risk of placebo effects influencing outcomes. This is particularly problematic in acupuncture trials where sham techniques may activate neuropathways, complicating the assessment of acupuncture’s true therapeutic effect.

To address these gaps, future research should prioritize the development of standardized treatment protocols for both acupuncture and HM. Larger, multi-center trials are needed to enhance the generalizability of findings and reduce bias. Furthermore, comprehensive subgroup analyses and mechanistic studies are essential to understanding the specific biological effects of these therapies. The inclusion of long-term outcome data, particularly regarding live births and infant health, would provide a more complete picture of the effectiveness and safety of EATM in ART.

### 4.5. Variability in Acupuncture and HM

The variability in acupuncture and HM treatments across studies significantly contributes to heterogeneity in EATM approaches:Different Acupuncture Techniques: Variations in types of acupuncture (e.g., body, electro-acupuncture, auricular), stimulation methods (manual vs. electrical), needle placement, and treatment duration introduce inconsistencies in administration, leading to variable outcomes across studies.Sham Acupuncture Impact: The range of sham techniques used affects control comparisons and complicates efficacy evaluation. Some sham methods may cause physiological responses, skewing results and making it difficult to assess the true effects of acupuncture.HM Diversity: The wide variety of herbal formulas (e.g., powders, pills, decoctions), dosages, and timing (before, during, or after IVF) complicates comparisons. These variations prevent clear conclusions about the efficacy of specific EATM interventions.

Overall, these differences contribute to the inconsistent findings, impacting the reliability and generalizability of the study results.

### 4.6. Practice and Future Research Implications

Future research should focus on addressing the variability in EATM protocols by standardizing treatment approaches according to STRICTA (Standards for Reporting Interventions in Clinical Trials of Acupuncture) criteria [65]. This includes protocol specifics, acupoint selection, session details, practitioner expertise, and standardized dosages of HM. Larger, multi-center RCTs with well-defined protocols are essential to reduce bias and improve generalizability. Studies should prioritize live birth outcomes and explore the physiological mechanisms by which EATM affects the reproductive system. Additionally, future trials should assess the comparative efficacy of different EATM modalities, like electro-acupuncture, manual acupuncture, and specific herbal formulations. Such research could provide more robust scientific evidence on the integration of EATM into ART, potentially offering more holistic treatment options for fertility.

### 4.7. The Potential Policy Implications

The findings of this study suggest actionable strategies for incorporating EATM into assisted ART protocols:Standardization of EATM Treatments in ART: Policymakers should develop unified guidelines for the application of acupuncture and HM in ART. This includes standardizing treatment protocols to improve consistency, reproducibility, and clinical outcomes across settings.Practitioner Training and Certification: Establishing specialized training and certification programs for EATM practitioners working in ART settings can ensure safe, high-quality, and evidence-informed care.Insurance and Accessibility: Expanding insurance coverage to include evidence-based EATM therapies can improve accessibility and reduce financial barriers for patients seeking integrative ART support.Interdisciplinary Collaboration: Fostering collaboration between reproductive endocrinologists and EATM practitioners may promote comprehensive and patient-centered care, leading to better clinical outcomes and patient satisfaction.

By addressing these areas, healthcare systems can create a framework for the safe and effective integration of EATM into reproductive care.

### 4.8. Integration of EATM in Clinical Practice

A collaborative model that integrates East Asian traditional medicine (EATM) with assisted reproductive technology (ART) may enhance patient outcomes across different phases of fertility treatment. Based on findings from 37 RCTs, the following framework outlines suggested uses of acupuncture and herbal medicine (HM):Preparation Phase (3 months before IVF)

Acupuncture and HM may help to regulate the menstrual cycle, balance hormones, and improve reproductive health.

Frequency: Acupuncture: 1–2 sessions/week; HM: 2–3 times/day.

2.Ovarian Stimulation to Egg Retrieval

Treatments aim to support ovarian response, promote follicle development, and reduce the side effects of stimulation drugs.

Frequency: Acupuncture: daily or every other day; HM: 2–3 times/day.

3.Embryo Transfer (ET) Day

Acupuncture is often administered before and after ET to enhance endometrial receptivity and reduce stress.

Frequency: Twice on ET day (pre- and post-transfer).

4.Post-Transfer to Beta-hCG Test

Continued treatment may support implantation and reduce early pregnancy loss.

Frequency: Acupuncture: every 1–2 days based on patient response.

5.Post-Pregnancy Confirmation (up to 12 weeks)

Acupuncture may help maintain early pregnancy and lower miscarriage risk.

Frequency: Once per week

For HM, prescriptions are tailored to the patient’s condition and ART phase, aiming to improve endometrial thickness, hormonal balance, and pregnancy outcomes. This model should be guided by evidence-based protocols and close collaboration between licensed EATM and ART practitioners to ensure safety and clinical effectiveness.

## 5. Conclusions

This meta-analysis provides compelling evidence that East Asian traditional medicine (EATM) interventions—particularly acupuncture and herbal medicine—can significantly improve reproductive outcomes, including clinical pregnancy and live birth rates, when used alongside assisted reproductive technology, without increasing adverse events or dropout rates. These findings support the potential safety and effectiveness of EATM as an adjunct to fertility treatment.

Subgroup analyses further reinforce the therapeutic benefits of true acupuncture over sham acupuncture and herbal medicine over placebos. However, the notable heterogeneity and variability in study quality highlight the need for further high-quality research.

To validate these findings and guide clinical integration, future studies should prioritize large-scale, multi-center, and double-blind randomized controlled trials with standardized treatment protocols and rigorous methodological designs. Mechanistic research and long-term outcome studies across diverse populations are also essential in order to establish the role of East Asian traditional medicine in reproductive medicine.

Our findings support the use of EATM as a complementary approach to enhance ART outcomes, while highlighting the need for standardized protocols and high-quality future research. Given the demonstrated effectiveness and safety of EATM, future health policies could consider supporting its integration into ART programs. This should include insurance coverage for evidence-based complementary therapies.

## Figures and Tables

**Figure 1 healthcare-13-01326-f001:**
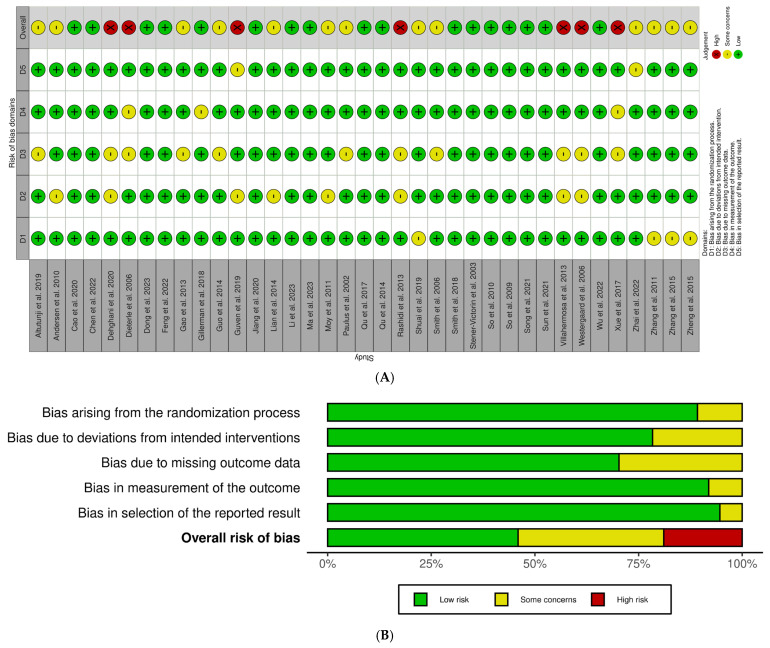
Traffic light plot for assessment of risk of bias of each included randomized trial (**A**) and weighted plot for assessment of overall risk of bias B using Cochrane RoB 2 tool [25,26] (**B**) (n = 37 studies). Traffic light plot reports five risk of bias domains: D1, bias arising from randomization process; D2, bias due to deviations from intended intervention; D3, bias due to missing outcome data; D4, bias in measurement of outcome; D5, bias in selection of reported result; green circle represents low risk of bias, yellow circle indicates some concerns of risk of bias, red circle reports high risk of bias.

**Figure 2 healthcare-13-01326-f002:**
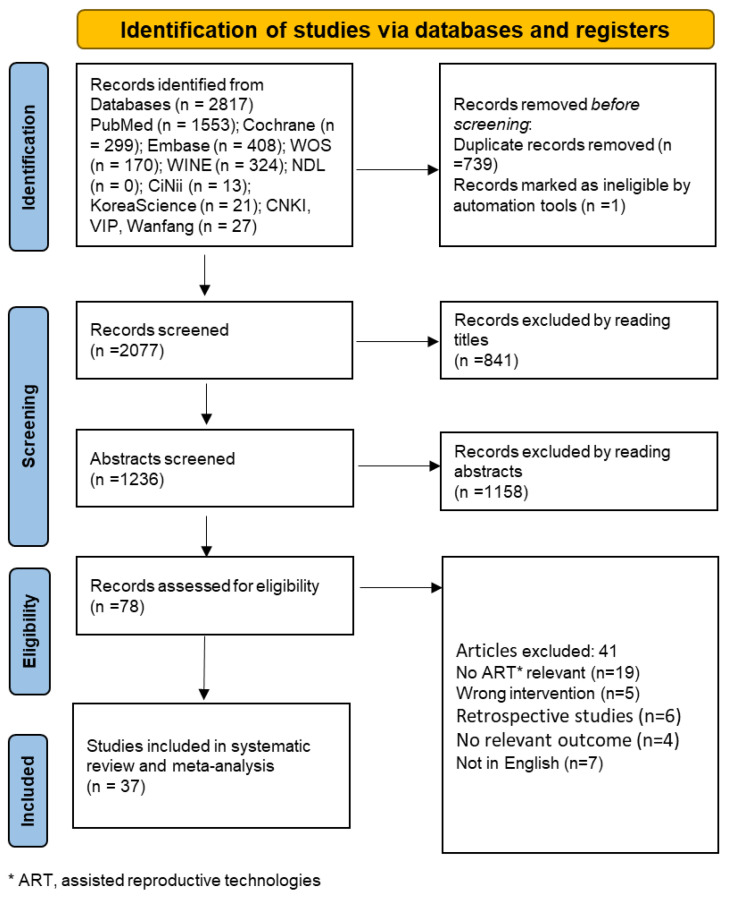
Flowchart of study selection.

**Figure 3 healthcare-13-01326-f003:**
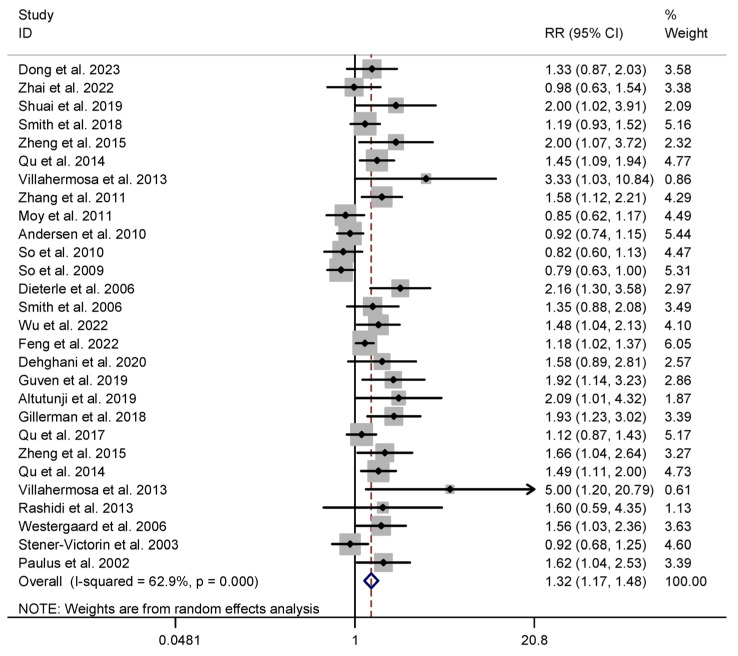
Meta-analysis of clinical pregnancy rate (acupuncture vs. control group), conducted using a random-effects model in Stata 12.0 [61].

**Figure 4 healthcare-13-01326-f004:**
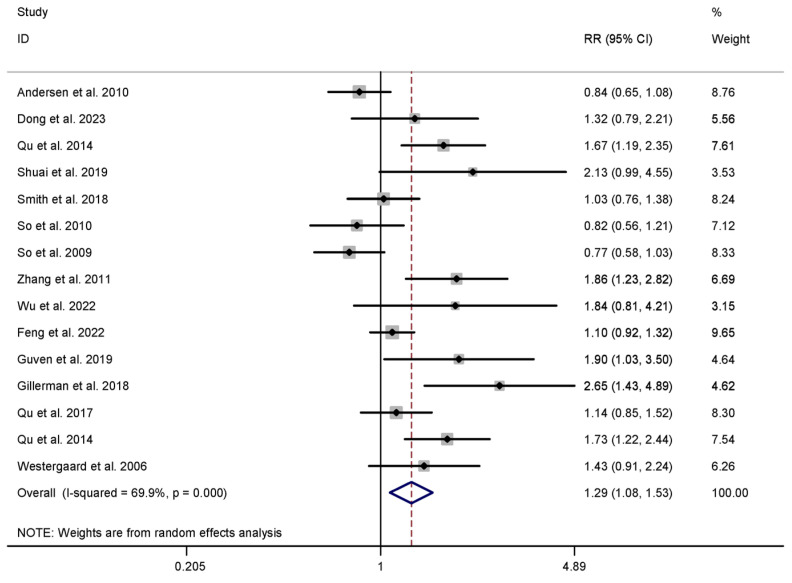
Meta-analysis of live birth rate (acupuncture vs. control group), conducted using a random-effects model in Stata 12.0 [61].

**Figure 5 healthcare-13-01326-f005:**
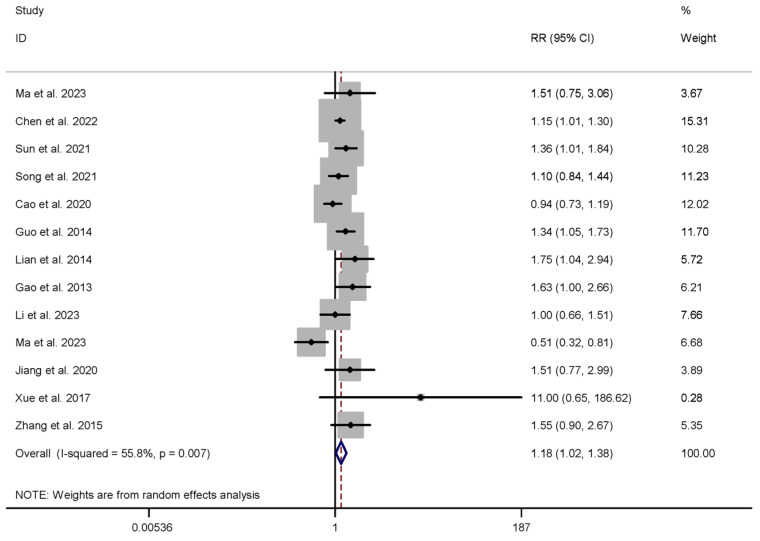
Meta-analysis of clinical pregnancy rate (herbal medicine vs. control group), conducted using a random-effects model in Stata 12.0 [61].

**Figure 6 healthcare-13-01326-f006:**
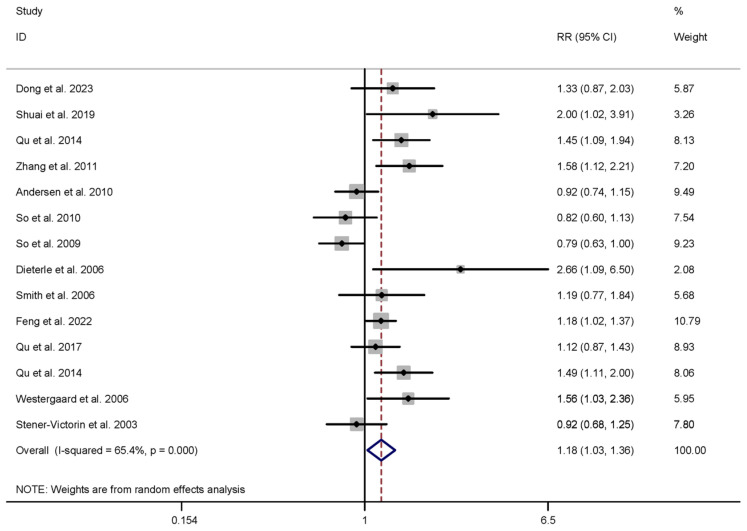
Meta-analysis of implantation rate (acupuncture vs. control group), conducted using a random-effects model in Stata 12.0 [61].

**Figure 7 healthcare-13-01326-f007:**
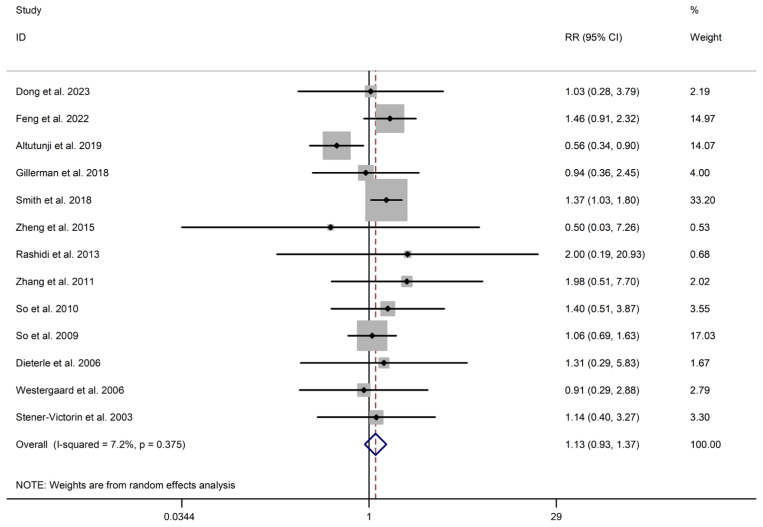
Meta-analysis of adverse events (EATM vs. control group), conducted using a random-effects model in Stata 12.0 [61].

**Figure 8 healthcare-13-01326-f008:**
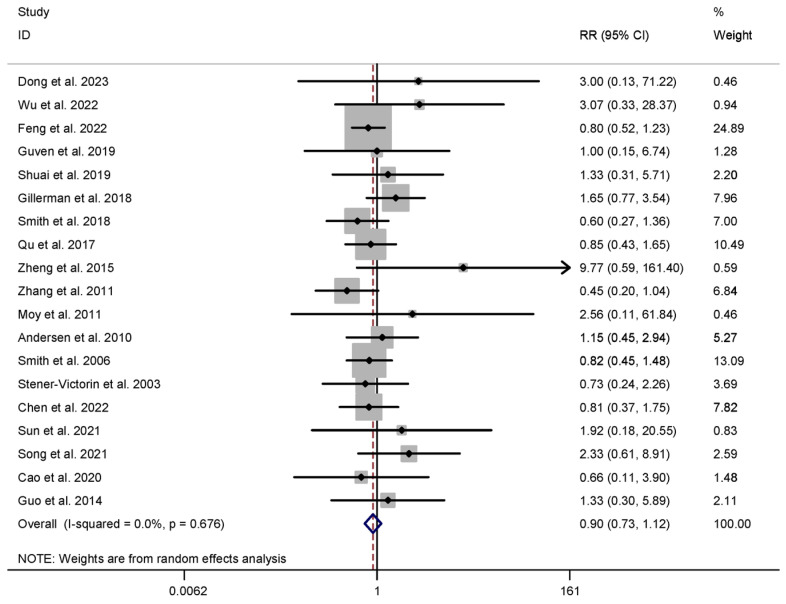
Meta-analysis of dropout rates (EATM vs. control group), conducted using a random-effects model in Stata 12.0 [61].

**Figure 9 healthcare-13-01326-f009:**
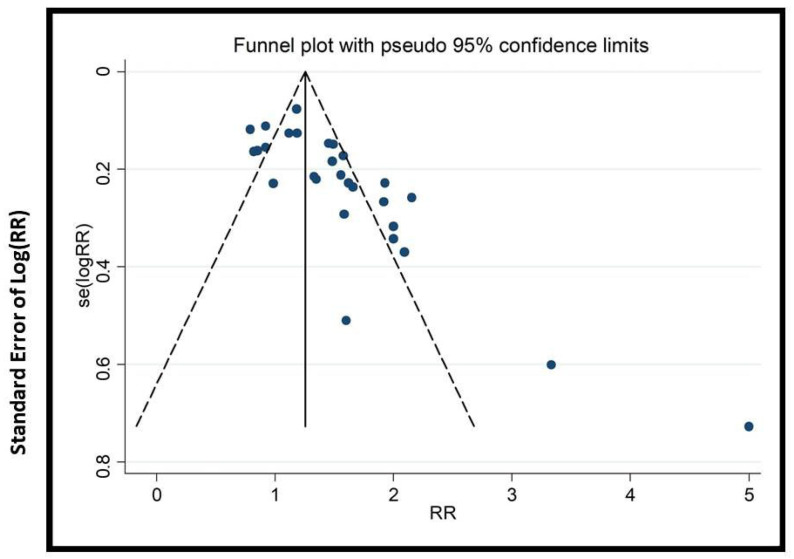
Funnel plot of clinical pregnancy rate (acupuncture vs. control group), conducted in Stata 12.0 [61].

**Figure 10 healthcare-13-01326-f010:**
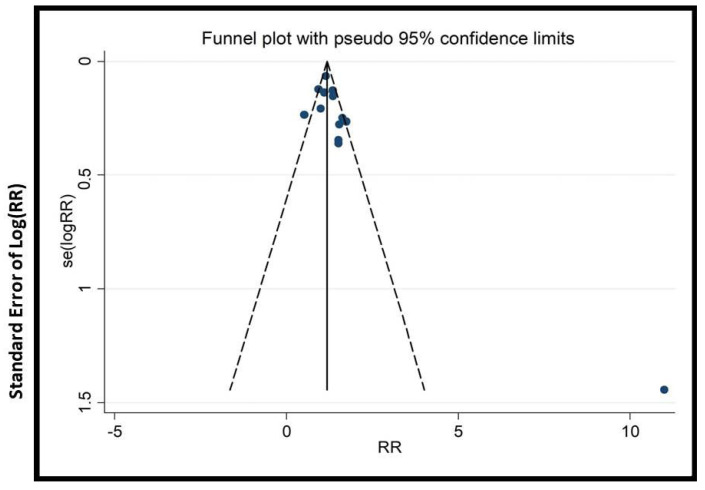
Funnel plot of clinical pregnancy rate (herbal medicine vs. control group), conducted in Stata 12.0 [61].

**Figure 11 healthcare-13-01326-f011:**
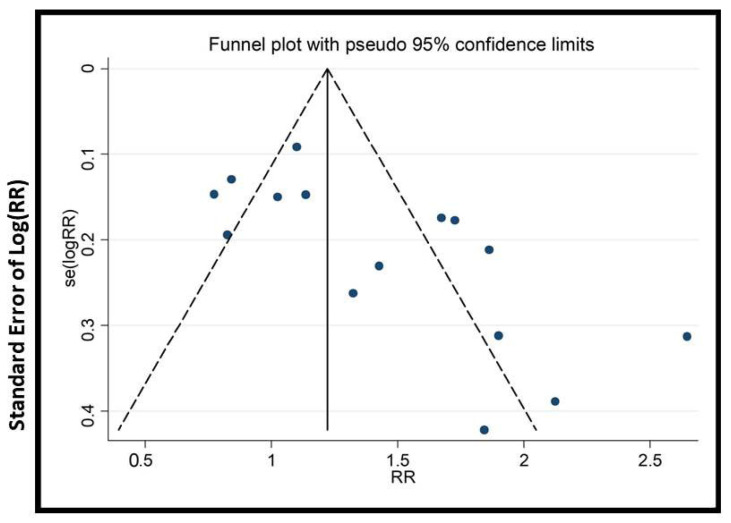
Funnel plot of live birth rate (acupuncture vs. control group), conducted in Stata 12.0 [61].

**Figure 12 healthcare-13-01326-f012:**
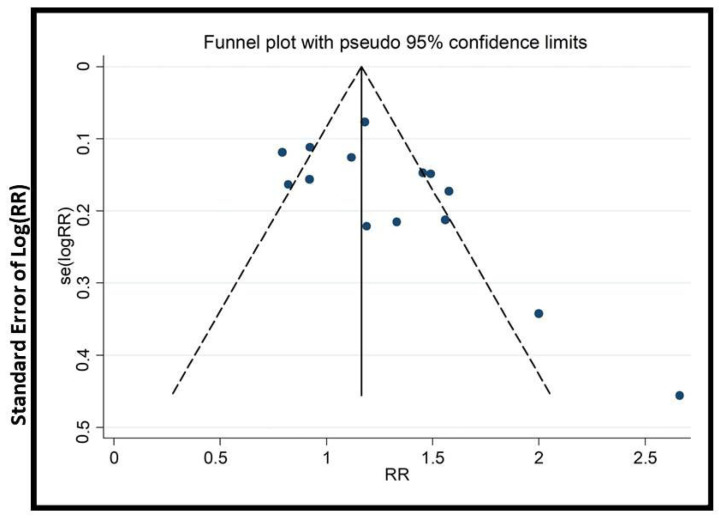
Funnel plot of implantation rate (acupuncture vs. control group), conducted in Stata 12.0 [61].

**Figure 13 healthcare-13-01326-f013:**
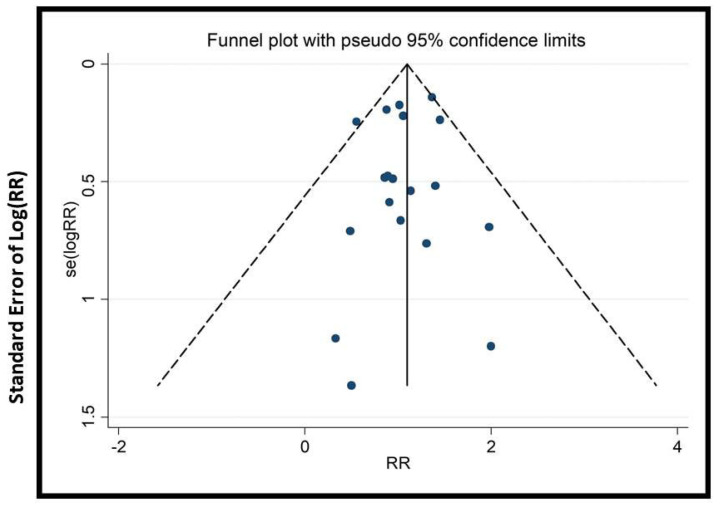
Funnel plot of adverse events (ETAM vs. control group), conducted in Stata 12.0 [61].

**Figure 14 healthcare-13-01326-f014:**
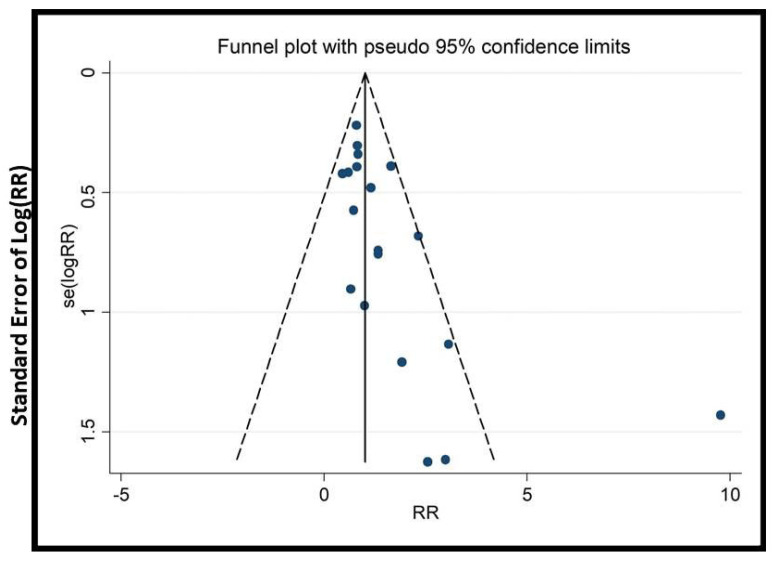
Funnel plot of dropout rates (ETAM vs. control group), conducted in Stata 12.0 [61].

**Figure 15 healthcare-13-01326-f015:**
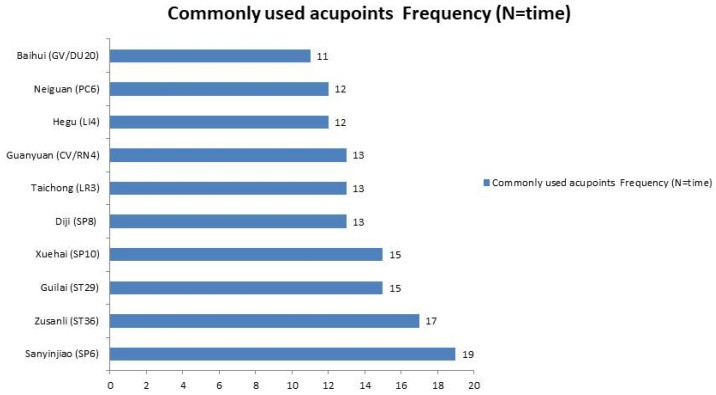
Commonly used acupoints.

**Figure 16 healthcare-13-01326-f016:**
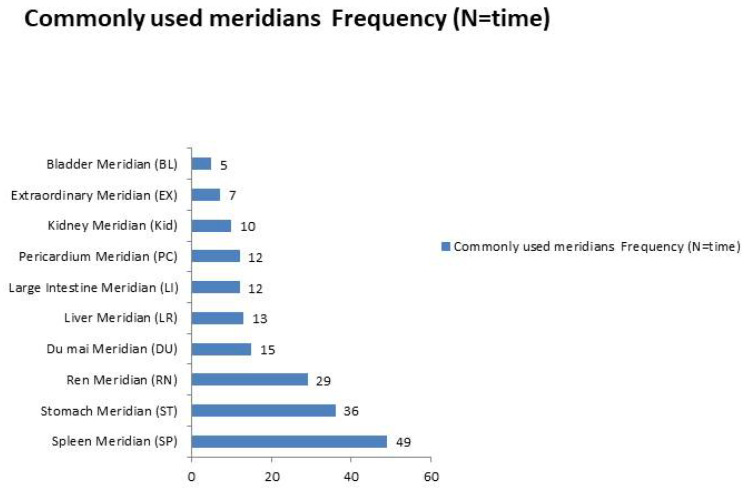
Commonly used meridians.

**Table 1 healthcare-13-01326-t001:** Meta-analysis results comparing intervention and control groups.

Outcomes	No. of Studies	No. of IG Patients	No. of CG Patients	RR (95% CI)	Study Heterogeneity		
					χ^2^	df	I^2^ (%)
Primary outcomes							
Acupuncture vs. Sham+ No intervention							
Clinical pregnancy rate	25	3565	3045	1.316 (1.171, 1.480)	72.68	27	62.9%
Live birth rate	14	2517	2096	1.287 (1.081, 1.533)	46.48	14	69.9%
Herbal medicine vs. Sham+ No intervention							
Clinical pregnancy rate	12	2142	2201	1.184 (1.017, 1.379)	27.14	12	55.8%
Live birth rate	3	1407	1411	1.147 (1.010, 1.303)	0.32	2	0.0%
Secondary outcomes							
Acupuncture vs. Sham+ No intervention							
Implantation rate	13	2302	1871	1.183 (1.028, 1.363)	37.57	13	65.4%
Adverse events *	13	1064	1092	1.125 (0.926, 1.367)	12.92	12	7.2%
Herbal medicine vs. Sham+ No intervention							
Implantation rate	5	1488	1547	1.106 (0.968, 1.264)	4.65	4	14.0%
Adverse events	6	771	706	0.916 (0.726, 1.157)	1.98	5	0.0%

* one trial was excluded from the subgroup analyzed due to the data being in a different form [11]. No., number; RR, risk ratio; CI, confidence interval; IG, intervention group; CG, control group.

**Table 2 healthcare-13-01326-t002:** Subgroup analyses of CAEPR, LBR, IR and AE (IG vs. CG).

Factor	Outcome or Subgroup	No. of Studies	No. of Patients	RR (95% CI)	I^2^	*p* Value
IG vs. CG CPR						
Type of CG	Sham Ac	14	3730	1.218 (1.019, 1.455)	69.7%	0.030
	No acupuncture	14	2880	1.416 (1.231, 1.629)	42.9%	0.000
	Placebo HM	8	3727	1.211 (1.071, 1.370)	32.2%	0.002
	No HM	5	616	1.101 (0.646, 1.876)	72.7%	0.724
LBR						
Type of CG	Sham Ac	8	2743	1.152 (0.892, 1.488)	74.4%	0.277
	No acupuncture	7	1870	1.465 (1.163, 1.846)	57.7%	0.001
	Placebo HM+ No HM	3	2818	1.147 (1.010, 1.303)	0.0%	0.035
IR						
Type of CG	Sham Ac	9	2317	1.192 (0.952, 1.493)	72.3%	0.125
	No acupuncture	5	1856	1.201 (1.031, 1.400)	41.6%	0.019
	Placebo HM+ No HM	5	3035	1.106 (0.968, 1.264)	14.0%	0.137
AE						
Type of CG	Sham Ac+ No acupuncture	13	2156	1.125 (0.926, 1.367)	7.2%	0.236
	Placebo HM+ No HM	6	1477	0.916 (0.726, 1.157)	0.0%	0.463

No., number; CG, control group; RR, risk ratio; CI, confidence interval; CPR, clinical pregnancy rate; LBR, live birth rate; IR, implantation rate; AE, adverse event; Ac, acupuncture; HM, herbal medicine. Note: The comparison groups in this table are clearly defined as the intervention group (IG) vs. the control group (CG), ensuring clarity in interpretation. The results reflect the effects of EATM interventions relative to different control conditions.

**Table 3 healthcare-13-01326-t003:** Sensitivity analysis comparing EATM with sham and control groups.

Outcomes	No. of Studies	Sample Size	RR	Effects Model	95% CI	I^2^ Value	Z Value	*p* Value
Vs. sham Ac/placebo HM								
LBR (Ac)	8	2743	1.152	Random	0.892, 1.488	74.4%	1.09	0.277
CPR (Ac)	14	3730	1.218	Random	1.019, 1.455	69.7%	2.17	0.030
CPR (HM)	8	3727	1.211	Random	1.071, 1.370	32.2%	3.04	0.002
Vs. CG								
LBR (Ac)	7	1870	1.465	Random	1.163, 1.846	57.7%	3.24	0.001
CPR (Ac)	14	2880	1.416	Random	1.231, 1.629	42.9%	4.88	0.000
CPR (HM)	5	616	1.101	Random	0.646, 1.876	72.7%	0.35	0.724

Ac, acupuncture; HM, herbal medicine; CI, confidence interval; RR, risk ratio; CPR, clinical pregnancy rate; LBR, live birth rate.

## Data Availability

Data are contained within the article and Appendix A: All data, including search strategy (Appendix A), study characteristics (Appendix A), Begg’s test and Egger’s test (Appendix A), GRADE profile (Appendix A), details of AE (Appendix A), reasons for dropping out (Appendix A), and commonly used acupoints and meridians (Appendix A) are available in the article and its online Appendix A.

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
