# Peer review of "Integrating Acupuncture and Herbal Medicine into Assisted Reproductive Technology: A Systematic Review and Meta-Analysis of East Asian Traditional Medicine"

_healthcare, 2025, doi:10.3390/healthcare13111326_

Round 1
Reviewer 1 Report
Comments and Suggestions for Authors
This is an interesting research article with adequate novelty. Some points should be addressed.
Lines 72–73
Referring back to the statement in the Introduction that "this systematic review and meta-analysis aim to address gaps in...", it would be valuable to add a paragraph discussing whether there are any expert opinions or clinical guidelines regarding the integration of East Asian Traditional Medicine into Assisted Reproductive Technology. A recent study has been published:
Li, X., Lee, H. W., Zhou, T., Robinson, N., Mio Hu, X. Y., Dunjić, M., Wang, F., Zhang, R., Panel, C., Zhu, Y., & Qu, F. (2025). Non-pharmacological interventions involving traditional Chinese medicine for assisted reproductive technology: A group consensus. Integrative Medicine Research, 14(2), 101137. https://doi.org/10.1016/j.imr.2025.101137
Lines 222 & 231
This information is already provided in the Abstract, which states that "data extraction and quality assessment were performed independently by two authors." Later, in section 2.7, the same point is unnecessarily repeated. In my opinion, this could be condensed into a single, concise sentence to avoid redundancy.
Lines 239 & 258
Similarly, in section 2.8, related information appears to be repeated. It may be worth considering merging both instances into a single, well-structured sentence to improve clarity and reduce redundancy.
The results are presented through a very large number of figures and tables. It might be helpful to consider condensing or streamlining them for clarity and ease of interpretation.
In the Discussion section, care should be taken to avoid repeating the results, especially since the authors begin the section by doing so.
Author Response
Dear Reviewer,
We sincerely thank you for your thoughtful review and constructive feedback on our manuscript. Your comments have been very helpful in improving the clarity, organization, and overall quality of our work. Below, we provide point-by-point responses to each of your suggestions.
Comment 1 – Expert Consensus on EATM Integration (Lines 72–73)
Reviewer’s comment:
Referring back to the statement in the Introduction that "this systematic review and meta-analysis aim to address gaps in...", it would be valuable to add a paragraph discussing whether there are any expert opinions or clinical guidelines regarding the integration of East Asian Traditional Medicine into Assisted Reproductive Technology. A recent study has been published:
Li, X., Lee, H. W., Zhou, T., Robinson, N., Mio Hu, X. Y., Dunjić, M., Wang, F., Zhang, R., Panel, C., Zhu, Y., & Qu, F. (2025). Non-pharmacological interventions involving traditional Chinese medicine for assisted reproductive technology: A group consensus. Integrative Medicine Research, 14(2), 101137. https://doi.org/10.1016/j.imr.2025.101137
Response:
Thank you for this valuable suggestion. We agree that including recent expert consensus enhances the clinical relevance of our study. We have added a paragraph to the Introduction discussing the 2025 consensus article by Li et al., which supports the integration of non-pharmacological TCM interventions into ART. This addition appears immediately after we present the study rationale.
Comment 2 – Redundancy in Section 2.7 (Lines 222 & 231)
Reviewer’s comment:
This information is already provided in the Abstract. Later, in Section 2.7, the same point is unnecessarily repeated.
Response:
Thank you for this helpful observation. We have revised Section 2.7 to present the information more concisely. The revised sentence reads: “Two authors independently conducted the study selection, data extraction, and quality assessment, with discrepancies resolved through discussion or by consulting a third reviewer.” This change removes redundancy while preserving clarity.
Comment 3 – Repetition in Section 2.8 (Lines 239 & 258)
Reviewer’s comment:
In Section 2.8, related information appears to be repeated. It may be worth considering merging both instances into a single, well-structured sentence.
Response:
Thank you for your suggestion. We have revised Section 2.8 to eliminate redundancy and improve clarity. The revised section combines methodological details into a concise format while maintaining all relevant content regarding the risk of bias assessment.
Comment 4 – Streamlining Figures and Tables
Reviewer’s comment:
The results are presented through a very large number of figures and tables. It might be helpful to consider condensing or streamlining them.
Response:
Thank you for pointing this out. We have condensed the presentation of our findings by moving Figures 15 and 16 to the supplementary materials in the appendix. However, we retained all forest plots in the main text to demonstrate that a full range of subgroup analyses was performed to assess the efficacy and safety of EATM interventions comprehensively.
Comment 5 – Avoiding Repetition in Discussion Section
Reviewer’s comment:
In the Discussion section, care should be taken to avoid repeating the results, especially since the authors begin the section by doing so.
Response:
Thank you for this insightful comment. In response, we have revised the opening paragraph of the Discussion section (Section 4.1) to begin with a summary interpretation of the findings, rather than repeating detailed statistical results. The numerical data now appears in the second paragraph. This restructuring improves the flow of the Discussion and aligns with its purpose of interpreting and contextualizing the results, rather than restating them.
We are grateful for your constructive comments, which have contributed significantly to improving the quality and clarity of our manuscript.
Sincerely,
Xiangping Peng & Gerhard Litscher (on behalf of all authors)
Corresponding Author

Reviewer 2 Report
Comments and Suggestions for Authors
The title is clear and accurately reflects the study's focus.
Consider specifying the types of East Asian Traditional Medicine (EATM) interventions (e.g., acupuncture, herbal medicine) for greater precision.
Abstract is Well-structured with clear objectives, methods, results, and conclusions.
The background could briefly mention the global prevalence of infertility to emphasize its significance.
The results section effectively summarizes key findings but could clarify the magnitude of improvements if available.
The conclusion is strong but could highlight the need for standardized protocols in future research.
Keywords are appropriate and cover the main themes. Consider adding "in vitro fertilization (IVF)" and "randomized controlled trials (RCTs)" for better indexing.
The introduction provides a solid foundation
In this section: Given the longstanding use of acupuncture and HM in reproductive health, these therapies have garnered attention for their potential to enhance pregnancy outcomes and live birth rates when combined with ART. Consequently, researchers have increasingly 66 focused on exploring adjunctive therapies in ART. Add related references. Also add some studies on the broader applications of EATM in infertility-related conditions (e.g., endometriosis, PCOS, male infertility, diabetes, male infertility and …) to strengthen the rationale. Some of these tretments results in higher embryo implantation and pregnancy.
It is suggested to use to below studies to complete this section:
Polycystic ovaries and herbal remedies: A systematic review
Comparing The Effects of Glycyrrhiza glabra Root Extract, A Cyclooxygenase-2 Inhibitor (Celecoxib) and A Gonadotropin-Releasing Hormone Analog (Diphereline) In A Rat Model of Endometriosis
The transition to ART limitations is smooth, but the link between EATM and ART could be more explicit earlier.
Write the objective of the study as below:
This systematic review and meta-analysis aimed to evaluate the efficacy and safety of East Asian Traditional Medicine (EATM), including acupuncture and herbal medicine (HM), as complementary therapies for improving clinical pregnancy rates (CPR) and live birth rates (LBR) in women undergoing assisted reproductive technology (ART). By analyzing randomized controlled trials (RCTs), the study sought to determine whether EATM interventions enhance ART outcomes compared to sham treatments, placebo, or standard care alone, while also assessing adverse events and heterogeneity across studies. Additionally, the review aimed to identify commonly used acupoints and herbal formulations, explore potential mechanisms of action, and highlight gaps in current evidence to guide future research and clinical integration.
This section is not necessary: Definitions of CPR, LBR, IR, and AE. Instead of explaining write an overall text and add reference.
Methods is Comprehensive and rigorous, adhering to PRISMA and PROSPERO guidelines.
The search strategy is detailed but could clarify why certain databases (e.g., Korean/Japanese) were included given the focus on EATM.
Inclusion/exclusion criteria are well-defined, but combining acupuncture and herbal medicine may introduce heterogeneity; justify this decision.
Results
Data presentation is systematic, with clear tables and figures.
The meta-analysis results are robust, Safety outcomes are well-reported.
Discussion is Strengths (large sample size, rigorous methodology) are appropriately highlighted.
The conclusion is supported by evidence but could emphasize policy recommendations (e.g., insurance coverage for EATM in ART).
Author Response
Dear Reviewer,
We sincerely thank you for your insightful and constructive comments on our manuscript. Your feedback has helped us improve the quality and clarity of our work. Please find below our detailed point-by-point responses to each of your comments.
Comment 1 – Title Specificity
Reviewer’s comment:
The title is clear and accurately reflects the study's focus. Consider specifying the types of East Asian Traditional Medicine (EATM) interventions (e.g., acupuncture, herbal medicine) for greater precision.
Response:
Thank you for your comment. We appreciate the suggestion to improve precision. The title has been revised to specifically mention the two primary intervention types assessed in this review: acupuncture and herbal medicine.
Revised Title:
“Integrating Acupuncture and Herbal Medicine into Assisted Reproductive Technology: A Systematic Review and Meta-Analysis of East Asian Traditional Medicine”
Comment 2 – Global Prevalence in Background
Reviewer’s comment:
The background could briefly mention the global prevalence of infertility to emphasize its significance.
Response:
Thank you for this suggestion. We would like to clarify that the global prevalence of infertility is already mentioned in the first sentence of the Introduction section:
“The infertility affects millions of couples worldwide, with an estimated 10–15% of reproductive-aged couples experiencing difficulties conceiving.”
We believe this adequately highlights the significance of the topic.
Comment 3 – Clarifying Magnitude of Improvement
Reviewer’s comment:
The results section effectively summarizes key findings but could clarify the magnitude of improvements if available.
Response:
Thank you for this comment. We would like to point out that the magnitude of improvements is already clearly reported in the Results section of the Abstract as follows:
“EATM significantly improved ART outcomes. Acupuncture increased clinical pregnancy rates (CPR: RR 1.316, 95% CI 1.171–1.480) and live birth rates (LBR: RR 1.287, 95% CI 1.081–1.533). HM also enhanced CPR (RR 1.184) and LBR (RR 1.147).”
Comment 4 – Policy Recommendations in Conclusion
Reviewer’s comment:
The conclusion is strong but could highlight the need for standardized protocols in future research.
Response:
Thank you for this helpful suggestion. We have revised the conclusion in the Abstract and main text to include a recommendation for developing standardized acupuncture and herbal medicine protocols in future research.
Comment 5 – Keywords
Reviewer’s comment:
Consider adding "in vitro fertilization (IVF)" and "randomized controlled trials (RCTs)" for better indexing.
Response:
Thank you for your suggestion. We have added the keywords “in vitro fertilization (IVF)” and “randomized controlled trials (RCTs)” to enhance indexing. This brings the total number of keywords from 7 to 9, which we believe is still appropriate given the interdisciplinary nature of the study.
Comment 6 – Broader Applications of EATM
Reviewer’s comment:
Add studies on broader applications of EATM (e.g., PCOS, endometriosis).
Response:
Thank you for your suggestion. However, we respectfully chose not to include studies related to broader infertility-related conditions for the following reasons:
1. Our current review focuses specifically on the role of EATM in enhancing ART outcomes (pregnancy and live birth rates).
2. We are planning a separate systematic review focused on EATM applications in conditions such as PCOS and endometriosis.
3. The suggested study involving Glycyrrhiza glabra root extract does not represent traditional multi-herb formulations typically used in clinical practice.
We believe this decision ensures focus and methodological clarity in the current review.
Comment 7 – Linking EATM and ART Earlier in Introduction
Reviewer’s comment:
The link between EATM and ART could be more explicit earlier.
Response:
Thank you for your comment. We would like to point out that this connection is already made in the tenth line of the Introduction:
“Given these challenges, interest in complementary therapies, particularly East Asian traditional medicine (EATM), has increased. EATM, which includes acupuncture and HM, has been practiced for centuries and offers alternative approaches to infertility...”
We believe this clearly establishes the relevance of EATM in the ART context.
Comment 8 – Study Objective Wording
Reviewer’s comment:
Write the objective of the study as below:
This systematic review and meta-analysis aimed to evaluate the efficacy and safety of East Asian Traditional Medicine (EATM), including acupuncture and herbal medicine (HM), as complementary therapies for improving clinical pregnancy rates (CPR) and live birth rates (LBR) in women undergoing assisted reproductive technology (ART). By analyzing randomized controlled trials (RCTs), the study sought to determine whether EATM interventions enhance ART outcomes compared to sham treatments, placebo, or standard care alone, while also assessing adverse events and heterogeneity across studies. Additionally, the review aimed to identify commonly used acupoints and herbal formulations, explore potential mechanisms of action, and highlight gaps in current evidence to guide future research and clinical integration.
Response:
Thank you for your thoughtful suggestion. We have adopted your proposed wording in full and replaced our original objective paragraph with the version you drafted. Your phrasing is clear, comprehensive, and well-aligned with the scope of our study.
Comment 9 – Definitions of CPR, LBR, IR, AE
Reviewer’s comment:
This section is not necessary. Instead of explaining, write an overall text and add reference.
Response:
Thank you for your suggestion. We respect your recommendation and have removed the definitions section from the manuscript as advised.
Comment 10 – Use of Korean and Japanese Databases
Reviewer’s comment:
Clarify why certain databases (e.g., Korean/Japanese) were included.
Response:
Thank you for your insightful comment. We included Korean and Japanese databases because EATM has a strong clinical and research tradition in these countries. Many high-quality RCTs on acupuncture and herbal medicine are published in Korean and Japanese literature, making these databases essential for a comprehensive and culturally representative review.
Comment 11 – Combining Acupuncture and HM
Reviewer’s comment:
Combining acupuncture and herbal medicine may introduce heterogeneity; justify this.
Response:
Thank you for raising this important point. Although combining both therapies could introduce heterogeneity, they are both core modalities of EATM and share a common theoretical foundation. To account for this, we conducted subgroup analyses for acupuncture and HM separately, allowing us to evaluate their individual effects while presenting a comprehensive view under the EATM framework.
Comment 12 – Policy Implications in Conclusion
Reviewer’s comment:
The conclusion could emphasize policy recommendations (e.g., insurance coverage for EATM in ART).
Response:
Thank you for this thoughtful recommendation. We have added a sentence at the end of the Conclusion to highlight potential policy implications:
“Given the demonstrated effectiveness and safety of EATM, future health policies could consider supporting its integration into ART programs, including insurance coverage for evidence-based complementary therapies.”
We are grateful for your valuable feedback, which has significantly contributed to improving our manuscript.
Sincerely,
Xiangping Peng & Gerhard Litscher (on behalf of all authors)
Corresponding Author

Reviewer 3 Report
Comments and Suggestions for Authors
The aim of this study was to evaluate the effectiveness of EATM in improving clinical pregnancy and live birth outcomes in women undergoing ART. The authors conclude that the acupuncture and herbal medicine increased clinical pregnancy rates and live birth rates, an also appear to be a safe and effective complementary therapy for improving ART outcomes. The study protocol was registered in the PROSPERO database in accordance PRISMA checklist. The authors presented search strategy, selection citeria and also justified the choice of a random effects model for statistical analysis, which is appropriate for significant clinical heterogeneity.
Overall, the study is well designed and described. However, some points need to be improved:
- to improve comprehension, the suggestion on how to interpret the data from section 3.4 "When the RR is greater than 1 and both CIs are entirely above 1, this indicates that the treatment group has a higher likelihood of achieving the desired outcome compared to the control group" should be moved to section 2.9.
- for figures 3-8, 9-14, provide explanations of abbreviations.
- Perhaps section 4.8 should be rephrased and shortened.
Author Response
Dear Reviewer,
We thank you for your careful evaluation of our manuscript and for your positive comments regarding the design, methodology, and clarity of our study. We also appreciate your constructive feedback, which has helped us refine our work. Please find our point-by-point responses below.
Comment 1 – Relocating Interpretation Guidance on RR and CI
Reviewer’s comment:
To improve comprehension, the suggestion on how to interpret the data from section 3.4 — "When the RR is greater than 1 and both CIs are entirely above 1, this indicates that the treatment group has a higher likelihood of achieving the desired outcome compared to the control group" — should be moved to section 2.9.
Response:
Thank you for this helpful suggestion. We agree that this interpretive guidance is better placed in the Methods section. Accordingly, we have moved this sentence from Section 3.4 to Section 2.8 (formerly Section 2.9) where we describe the statistical analysis approach.
Comment 2 – Clarifying Abbreviations in Figures
Reviewer’s comment:
For Figures 3–8 and 9–14, provide explanations of abbreviations.
Response:
Thank you for your comment. We have revised the figure legends for Figures 3–8 and 9–14 to include full explanations of all abbreviations used. This change enhances clarity for readers.
Comment 3 – Rephrasing and Shortening Section 4.8
Reviewer’s comment:
Perhaps section 4.8 should be rephrased and shortened.
Response:
Thank you for your valuable suggestion. In response, we have rephrased and condensed Section 4.8 to improve clarity and focus. The revised version reduces the word count from 299 to approximately 221 words while preserving the essential clinical recommendations. We streamlined the language and structure to make it more concise, while maintaining the practical guidance regarding the integration of acupuncture and herbal medicine across ART phases.
We are grateful for your thoughtful feedback and constructive input, which have helped enhance the overall quality of our manuscript.
Sincerely,
Xiangping Peng & Gerhard Litscher (on behalf of all authors)
Corresponding Author
